# A Comprehensive Evaluation System for the Stabilization Effect of Heavy Metal-Contaminated Soil Based on Analytic Hierarchy Process

**DOI:** 10.3390/ijerph192215296

**Published:** 2022-11-19

**Authors:** Suxin Zhang, Cheng Hu, Jiemin Cheng

**Affiliations:** 1College of Geography and Environment, Shandong Normal University, Jinan 250358, China; 2School of Mathematics and Statistics, Shandong Normal University, Jinan 250358, China

**Keywords:** evaluation indicator, evaluation model, AHP, stabilization remediation, heavy metals, farmland soil

## Abstract

Stabilization technology is widely used in the remediation of heavy metal-contaminated farmland soil. However, the evaluation method for the remediation effect is not satisfactory. To scientifically evaluate the remediation effect, this study constructed a comprehensive evaluation system by bibliometric analysis and an analytic hierarchy process (AHP). Ultimately, 16 indicators were selected from three aspects of the soil, crops, and amendment. The 16 indicators are divided into three groups, namely indicators I that can be evaluated according to the national standards of China, indicators II that can be evaluated according to the classification management of farmland and Indicators III that are the dynamic change indicators without an evaluation criterion. Comprehensive scores for 16 indicators were calculated using three response models, respectively. According to the difference between the scores before and after the remediation, the remediation effect is divided into five levels, which are excellent, good, qualified, poor, and very poor. This study provides a theoretical basis and insightful information for a farmland pollution remediation and a sustainable utilization.

## 1. Introduction

Due to the combustion of some heavy fuels, such as heavy oil, leaded gasoline, waste-to-energy and fossil fuels, heavy metals are released into the environment during energy use in the form of flue gas or slag [1]. These heavy metals suffuse in the atmosphere, water and soil and cause various types of environmental pollution. Soil is the ultimate receptor for heavy metal pollutants. The survey results show that the contaminated land area in China accounts for 19.4% of the total cultivated land area, mainly due to heavy metal pollution, with Cd being the predominant contaminant [2]. The output and quality of agricultural goods are reduced when heavy metals enter the soil [3], and they can even enter the human body through the food chain [4,5], posing a threat to human health. Heavy metals will strongly interact with different proteins and enzymes once they enter the human body, leaving them inert [6]. They may also be abundant in specific human organs [7].

Recently, soil remediation has gotten a lot of attention because of issues with food safety and soil degradation caused by a heavy metal contamination [8]. The commonly used amendments are lime [9], calcium carbonate [10], fly ash [11], hydroxyapatite [12], biochar [13] and zeolite [14]. Studies have shown that various amendments exhibit different stabilization effects [10]. According to a review of the literature, China’s remediation of Cd-contaminated soils resulted in an available Cd reduction ranging from 1.06% to 91.00% [15,16]. In addition, the remediation effect could also be significantly influenced by the soil type, the crop species, the mechanisms of the amendments and the agricultural practices [17,18]. Therefore, the remediation effect of the amendments needs to be scientifically assessed before it is used on a large scale.

From the current stage of stabilization, the evaluation methods are divided into the following main areas. First, comparing the available heavy metals and the reduction in the heavy metal content in plants [19]. However, this method necessitates the development of a critical value standard for the available heavy metals in soil, which has not yet been established [20]. Second, there are approaches that use plant and microbiological indicators [21], but these techniques are vulnerable to weather, drought, pests and diseases, farming practices and other variables. Third, a comprehensive evaluation system was developed. For example, the evaluation system constructed by Zhao et al. [19] did not involve multiple heavy metals, and the scientific rationale used to grade the results and assign points to indicators was insufficient. Additionally, the Soil Environmental Quality Risk Control Standards for Soil Contamination on Agricultural Land (for a trial implementation) (GB 15618-2018) [22] also contain the screening and control values for the total heavy metals. However, the available heavy metals are altered by stabilization technology, so the standard cannot be used to assess the impact of stabilization on agricultural land.

The analytic hierarchy process (AHP) is one of the multi-criteria decision-making methods, which could be used to quantitatively and qualitatively judge the merits of the indicators [23]. Therefore it is suitable for evaluating the stabilization effect of contaminated soil. Additionally, it also works well with a small amount of data to demonstrate the completeness of the evaluation objectives [24]. To the authors’ knowledge, there are few published studies on the comprehensive remediation effects of heavy metal-contaminated farmland soils to address the above challenges.

The secret to a successful evaluation of the stabilization effects is the use of scientific and effective evaluation indicators. In addition, the evaluation standards and evaluation methods regarding the stabilization effect are lacking. Hence, this study intends to develop a comprehensive evaluation system consisting of the concerns on the soil, crop and amendment to determine the remediation sustainability of amendments in heavy metal-contaminated soil. A literature analysis, the Delphi method and the analytic hierarchy process were adopted to select the indicators of evaluating the impact of the amendment. This study may provide a scientific method or new ideas for the assessment of the stabilization effect in heavy metal-contaminated soil.

## 2. Research Methods

The study was completed in four steps. The first step is to identify the indicators for evaluating the stabilization effect of heavy metal-contaminated soil, the second step is to analyze and calculate the weights of the indicators, the third step is to specify the evaluation criteria and evaluation models for the indicators and the fourth step is for the laboratory application. The specific steps are shown in Figure 1.

### 2.1. Literature Analysis Method

The literature analysis approach [23] refers to the investigation of the gathered material to ascertain the character and state of the research subject and to derive one’s viewpoint. The high-rate indicators involved in this study were retrieved and counted from 424 papers published in the last 20 years about the stabilization remediation in heavy metal-contaminated soil (Figure 2). Based on what was present in the soil–plant system, these indicators were divided into three groups: soil, crop and amendment. Then, these indicators were further eliminated or optimized with the guidance of a theoretical analysis and expert advice.

### 2.2. Delphi Method

With the benefits of feedback, confidentiality and statistics, the Delphi method [25] solicits the opinions of numerous experts in various domains and seeks an expert’s consensus to resolve complex management challenges. Twenty experts in the domains of agrology, the remediation of a heavy metal contamination, environmental monitoring, environmental assessment and other related research fields were consulted for this study through the completion of a questionnaire.

### 2.3. Analytic Hierarchy Process

The analytic hierarchy process [24] is to decompose the decision problem into a target layer, criterion layer and indicator layer (scheme layer), solve the eigenvectors of the judgment matrix and the priority weight of each element to an element in the previous level and finally find the final weight of each indicator to the total target.

The specific calculation steps are as follows:

#### 2.3.1. Construction of Judgment Matrix

Construct a judgment matrix of two comparisons for each factor in the same layer. Assume that the evaluation target is *A* and the set of its lower-level evaluation indicators is *B = {b*_1_*, b*_2_*,..., b_n_}*, and construct the judgment matrix *P(A − B)*.
(1)$$P(A-B)=\left[\begin{array}{ccc} b_{11} & \cdots & b_{1n}\\ \vdots & \ddots & \vdots \\ b_{n1} & \cdots & b_{nn} \end{array}\right]$$
where *b_ij_* is the relative importance of factor *i* for *j* (*i*, *j* = 1, 2,..., n)

The Satty 1–9 scale was employed to determine the relative importance of factors *i* and *j*. (Table 1).

#### 2.3.2. Calculate the Maximum Eigenvalue of the Judgement Matrix Using the Sum-Product Method [26]

First, the judgement matrix P is regularized according to Equation (2).
(2)$${\hat{b}}_{ij}=\frac{b_{ij}}{\Sigma_{i=1}^{n}b_{ij}}$$
where *b_ij_* is the importance scale value of each element in the judgment matrix *P*, and n is the order of each judgment matrix (*n* = 1, 2,..., *n*). The same applies below.

Then, the judgment matrix is normalized according to Equations (3)–(5), and then the maximum characteristic root of the judgment matrix is calculated according to Equation (6).
(3)$${\bar{W}}_{i}=\Sigma_{j=1}^{n}{\hat{b}}_{ij}$$
(4)$$\bar{W}=\left[{\bar{W}}_{1},\;{\bar{W}}_{2},\ldots ,\;{\bar{W}}_{n}\right]$$
(5)$$W_{i}=\frac{{\bar{W}}_{i}}{\sum_{i=1}^{n}{\bar{W}}_{i}}$$
(6)$$\lambda_{max}=\sum_{i=1}^{n}\frac{{\left(AW\right)}_{i}}{nW_{i}}$$
where ${\bar{W}}_{i}$ is the sum of the elements of each row of the judgment matrix; $\bar{W}$ is the matrix composed of ${\bar{W}}_{i}$; $W_{i}$ is the corresponding element of the matrix ${\bar{W}}_{i}$ after the normalization; and $\lambda_{max}$ is the maximum characteristic root of the judgment matrix.

#### 2.3.3. Consistency Test

Among them, *CI* is the consistency indicator, *CR* is the consistency ratio, *RI* is the random consistency indicator and the specific value of RI is shown in Table 2.
(7)$$CI=\frac{\lambda_{max}-n}{n-1}$$
(8)$$CR=\frac{CI}{RI}$$

During the calculation of the weights, when *CR* < 0.1, the degree of the inconsistency of the judgment matrix is considered to be within the tolerable range, representing no logical errors in the important judgments of the indicators, and, conversely, the judgment matrix needs to be readjusted until the consistency condition is satisfied [27].

## 3. Results

### 3.1. Evaluation Principles

Contrary to the remediation of heavy metal-contaminated industrial soils, the focus of an agricultural soil remediation should not only be on the reduction in the available heavy metals but also on whether the farmland can be reclaimed. Therefore, the selection of the indicators should conform to the following principles: Comprehensive. The selection of the evaluation of the indicators should reflect the changes in all aspects of the soil–plant system.Objectivity. The selected indicators should reflect the remediation effect to the greatest extent possible, of which the quantitative analysis indicators should be the main focus.Stability. Highly stable indicators should be selected to ensure the relative stability of the evaluation results.Ease of evaluation. The indicators used to evaluate the stabilization effect should be easily measured or measurable with the available technical means, so that a numerical transformation and statistics can be performed during the evaluation process.

### 3.2. Evaluation Index Selection Analysis

#### 3.2.1. Soil Indicators

(1)Fertility Indicators

The soil’s physicochemical properties reflect the quality of the soil and influence the soil’s fertility. The soil’s pH value affects the effective release of the soil’s nutrients, the biological effectiveness of the heavy metals and the growth and development of plants [28]. In addition, a proper pH value can improve the soil’s quality by promoting microbial activity in the soil, and it has an important position in the evaluation of blunt remediation. The soil organic matter (SOM) refers to all forms of carbonaceous organic matter in soil, which is an important component of the soil’s fertility and directly reflects the soil’s fertility [25]. Additionally, what is known as a “soil nutrient reservoir” plays an important role in ensuring normal plant growth. If the SOM changes significantly after remediation, it will change the soil’s fertility and affect the soil’s growth capacity. The cation exchange capacity (CEC) represents the level of fertility of the soil [29] and can be used as an indicator to evaluate the fertility of the soil, while the CEC is of a great significance for studying the environmental behaviors of pollutants. Nutrients in soil, such as nitrogen, phosphorus and potassium, provide a large number of nutrients for a plant’s growth, are closely related to a crop’s physiological metabolism, growth and development and they yield its formation. They are the basis of soil fertility [30] and are indispensable elements for normal plant growth [31].

(2)Heavy metal indicators

The screening and control values for the total heavy metal (THM) are specified in the Chinese agricultural soil standard [22], so it should be an evaluation indicator for a stabilization remediation. Available heavy metal (AHM) refers to the fraction that is easily released into the soil and is more active, which is also absorbed by crops during their growth. However, it was found that the content of the AHMs in soil does not depend entirely on the THM, for example, in areas with a high THM, a high accumulation of heavy metal was not found in crops, but in some fields with a low THM, the toxic effect of heavy metals was obvious. Therefore, some experts proposed to use the AHMs to evaluate the toxicity of heavy metals. The purpose of stabilization is to reduce the biological effectiveness of heavy metals in soil [17], then this part should be the top priority in evaluating the merits of a stabilization remediation. In addition, the study of the morphology of heavy metals is essentially to observe the part of the soil that can be absorbed by organisms, which is the same purpose as the study of the AHMs.

(3)Microbiological indicators

The application of the amendment affects the microorganisms in the soil [32] due to the various structures and number of microorganisms in the soils and the sensitivity of the different functional enzymes to heavy metals [33]. This means that microbiological indicators also have an important role to play in the evaluation process. However, the determination of the properties of the soil’s enzymes and microorganisms is expensive and complex due to their operation. In addition, there are no mature evaluation norms for the nature of microorganisms in soil, so these three indicators were not included in the system in this study.

#### 3.2.2. Crop Indicators

The growth of crops can introduce heavy metals from the soil into the body, enriching them in the human body through the food chain, which is detrimental to human health [34]. Studies on the accumulation of heavy metals by crops have mainly focused on that which is above-ground, the edible parts and the roots. The level of heavy metal content in plants can reflect the available heavy metals in soil [35], which in turn can reflect the effect of stabilization. The biomass represents the health of the crop during the stabilization process. In the agricultural production process, production is the main concern of farmers, and if the production decreases after the stabilization, it will increase the economic pressure of farmers, so the evaluation of the production is essential.

#### 3.2.3. Amendment Indicators

The amendment cannot be ignored, even though the literature review reveals that relatively few researchers are concerned about its effects on the soil ecosystem. If the amendment contains more heavy metals, it can cause an exogenous introduction of heavy metal, making a stabilization remediation counterproductive. In addition, considering the ease of measurement and evaluation of the indicators, only the heavy metal in the amendment, cost and stability were chosen.

### 3.3. Evaluation Index System Determination

Based on the bibliometric analysis, combined with the principles for the selection of an indicator and a consultation with the experts, the evaluation system which was finally constructed is shown in Figure 3. It contains 1 target layer, 3 criterion layers and 16 indicator layers, and the 3 criterion layers are the soil criterion layer, crop criterion layer and amendment criterion layer. The soil criterion layer contains eight indicators, namely the THM, AHM, pH, SOM, CEC, available nitrogen (A–N), available phosphorus (A–P) and available potassium (A–K), which represent the change in the soil’s quality by a stabilization remediation; the crop criterion layer contains five indicators: the heavy metals in the above-ground parts, heavy metals in the roots, heavy metals in the edible part, biomass and yield, which represent whether the application of the amendment will improve the crop’s quality; and the amendment criterion layer consists of three indicators: the cost, heavy metals in the amendment and stability, which means whether the applied amendment is safe for production and judging the economic benefits.

### 3.4. Weight Analysis

According to the steps described in 2.3, the results of 20 questionnaires were calculated and analyzed, and the weighted average was taken to obtain the relative weights and combined weights of each indicator, as shown in Table 3. The results showed that the soil (0.544) and crops (0.316) accounted for significantly higher weights than the amendment (0.140) in the evaluation system for the stabilization effect of heavy metal-contaminated soil, and researchers paid more attention to the soil and crops when evaluating them. Under the soil fertility hierarchy, the pH (0.052) value has the greatest weight. Changes in the pH can cause changes in the soil’s properties. For example, a lower pH and soil acidification can lead to the activation of heavy metals in the soil and an increased toxicity. In the whole indicator layer, the AHMs (0.300) take up the largest weight. The purpose of a stabilization is to reduce the AHMs, and this part dominates the stabilization effect. In China’s current national standards for heavy metals, there are no provisions on the AHMs. Therefore, this part is a challenge, and there is an urgent need to establish a scientific and reasonable evaluation model and evaluation guidelines for the AHMs. Then, heavy metals in the edible parts (0.150) have the second highest weight, which is directly related to food security and needs great attention. In general, the calculation of the weight of each indicator is consistent with the content and focus considered in the evaluation of soil heavy metal pollution.

### 3.5. Classification of Indicators and Evaluation Standards

In a remediation project, it is necessary to assign evaluation criteria to each indicator to determine whether the remediation requirements are satisfied. Some of the indicators mentioned can be evaluated according to national standards of China, some only can be evaluated according to the classification management of farmland and some indicators have no evaluation criteria. Therefore, the indicators in the evaluation system are divided into the following three groups.

#### 3.5.1. Indicators I

Only the soil’s THMs, heavy metals in the edible parts and heavy metals in the amendment have been clearly specified in the national standards of China. These indicators should be strictly enforced because they have standardized evaluation methods and procedures that must be followed. These three indicators must meet the standards listed in Table 4.

#### 3.5.2. Indicators II

In addition to reducing the AHMs during the stabilization remediation, the soil properties and crop growth should also be expected to be no less than the original soil, or even better than the original soil.

With the Environmental Quality of Green Food Producing Areas (NY/T391-2013), the Quality Grades of Cropland (GBT 33469-2016) and the second national census nutrient grading standards [36,37], indicators such as the pH, CEC, SOM, A-N, A-P, A-K, biomass, production and cost are graded, although there are no clear requirements for them. As a result, they can be evaluated according to the classification management of farmland. Table 5 provides an explanation of the grading standards.

#### 3.5.3. Indicator III

The indicators of the AHMs, the heavy metals in the above-ground parts and in the roots and the stability are dynamic change indicators without an evaluation criterion. Therefore, their evaluation criteria can refer to previous research.

The accumulation of heavy metals by crops can be expressed in terms of the bio-accumulation factor (BAF) [38], which is calculated as follows.
(9)$$BAF=\frac{{\left[X\right]}_{crop}}{{\left[X\right]}_{soil}}\times 100\%$$
${\left[X\right]}_{crop}$ is the content of heavy metals in the crop, and ${\left[X\right]}_{soil}$ is the content of heavy metals in the soil.

The BAF can indicate the uptake of the elements by plants and, according to previous studies, can be divided into four classes: a BAF > 100% is a strong uptake, 10% < a BAF ≤ 100% is a moderate uptake, 1% < a BAF ≤ 10% is a weak uptake and a BAF < 1% is a very weak uptake [39,40]. Therefore, the accumulation of heavy metals by crops can be determined according to the difference in the BAF, which can also reflect the stabilization effect.

Most researchers use the reduction in the AHMs to evaluate the stabilization effect, and the reduction in the AHMs varies considerably between the amendments. When the types of amendments and their stabilization effects were counted, it was found that the reduction rate of cadmium in soil was as high as 99% and as low as 1.06%. The common amendments were, among others, lime, calcium-magnesium phosphate fertilizer, hydroxyapatite, seafoam, zeolite and biochar, and their reduction rates were able to reach 84.40%, 70.06%, 52.40%, 78.00%, 90.65% and 59.13%, respectively. It can be seen that the efficiency of conventional amendments is almost above 50% and can even reach 70%. Using the equidistance method, the reduction rate of the AHMs was classified into four grades: 0 < α < 25% as poor, 25% ≤ α < 50% as a medium, 50% ≤ α < 75% as good and 75% ≤ α < 100% as excellent. The reduction rate for the AHMs is calculated as follows.
(10)$$\alpha =\frac{X_{b}-X_{a}}{X_{b}}\times 100\%$$
$X_{b}$ is the AHMs before stabilization and $X_{a}$ is the AHMs after stabilization.

Amendments have been shown to maintain their stabilization effect for 3–5 years after their application to the soil [41]. In this study, the life cycle of 3 years was used as the criterion for evaluation.

### 3.6. Comprehensive Evaluation Model

The stabilization effect evaluation models are significantly different from the more mature soil pollution status evaluation models, which are relatively simple and mostly static. These indicators have different types and evaluation criteria. Therefore, the stabilization effect cannot be evaluated by just one model.

In order to eliminate the effects of different units and magnitudes of variation, the indicators should be normalized by referring to the standardized scoring function model used by Wang et al. [42] to obtain a standardized score from 0 to 1. Three different response curve models, “S” type, inverted “S” type and parabolic or midpoint type, are listed according to the requirements of the stabilization, and the response curve model is simplified to obtain the standardized scoring function as shown in Figure 4. The formulae for each scoring function and the applicable indicators are listed in Table 6. The pH value is too high or too low to affect the soil’s health, so a “parabolic or midpoint type” scoring function is used for the calculation. The higher the value of these indicators, such as the SOM, CEC, A-N, A-P, A-K, production, stability and biomass, the more consistent they are with the concept of a stabilization remediation, so an “S” type function is used (the higher the value, the higher the score). For the AHMs, THMs, the heavy metals in the edible part, above-ground parts, roots and amendment and the cost, it is generally accepted that the lower the heavy metal content, the lower the risk of a contamination and as the remediation effect gets higher, an inverted “S” type function is used (the lower the value, the higher the score). The threshold values L, H, L_1_ and H_1_ for each criterion function in Table 6 are set with reference to the evaluation criteria listed in 3.5.

A comprehensive weighted scoring model was constructed for the whole evaluation system [43], and the stabilization effect was judged by comparing the scores obtained before and after the stabilization.
(11)$$S_{i}=\sum_{i=1}^{n}\omega_{i}X_{i}$$
where $X_{i}$ is the normalized value of the indicator and $\omega_{i}$ is the weight (the weights of each indicator can be found in Table 4), $S_{i}$ is the comprehensive score of the entire evaluation system.

The target tier assessment score $S_{i}$ is graded according to the equidistance method, as shown in Table 7.

The score before the stabilization (S_b_) and the score after the stabilization (S_a_) were calculated separately, and S_b_ and S_a_ were compared according to Table 8 to obtain the final score S.

### 3.7. Laboratory Applications

The pot experiment was used to apply the evaluation system. The soil was collected from the farm of Shandong Normal University, mainly in the top layer (0–20 cm). The physicochemical properties of the soils are shown in Table 9. 

A simulated contaminated soil sample of 5 mg·Kg^−1^ Cd^2+^ was weighed out to 1.5 kg, respectively, and put into pots. A total of 1% reed biochar (RBC) and 1% hydroxyapatite (HAP) were added, respectively, and the treatment without an amendment was used as the control; 10 ryegrass plants were planted in each pot, and each treatment was repeated three times. In addition, 1.317 g of potassium phosphate and 0.6345 g of urea were added to each pot to keep the base fertilizer consistent. During the incubation, watering was done to maintain the soil’s moisture at 65% to 70% of the saturated water holding capacity in the field. The evaluation indicator was measured after 70 days. The actual values of the indicators and their scores are shown in Table 10.

The comprehensive scores S_i_ for the two amendments are shown in Table 10, where S_RBC_ is 0.371 and S_HAP_ is 0.471. The HAP was able to better enhance the quality of the nutrients in the soil, while the RBC was able to significantly increase the content of organic matter in the soil in the soil’s fertility. Meanwhile, both were good at promoting the growth of ryegrass and inhibiting the accumulation of Cd in the plant. This may be related to the large amount of A-P to the soil from the HAP. In addition, the application of biochar has been reported to not only adjust the soil’s structure, but also increase the soil’s organic matter and promote the plant’s growth. The HAP was better than the RBC in reducing the available Cd. This result was similar to the one obtained by Zhang et al. [44], who found that the application of hydroxyapatite and biochar can significantly reduce Cd by 71.83% and 57.28%, respectively. According to the equidistant grading method in Table 7, the RBC has a grade IV and the HPA has a grade III. No amendments were applied to the soil in the CK, and its score represents the score of the soil before stabilization with a grade of V. According to the evaluation method in Table 8, the S of the RBC is one, the soil quality after the stabilization improved by one grade, the evaluation result is good, the S of the HAP is two, the soil quality after the stabilization improved by two grades and the evaluation result is excellent. On the whole, the stabilization effect of the HAP is better than that of the RBC. The results show that the evaluation system and evaluation model are feasible.

## 4. Discussion

### 4.1. Evaluation Index

The scientific selection of the representative indicators is the key to the evaluation of the stabilization effect. However, too many evaluation indicators will make the evaluation session tedious and complicated, which is not conducive to the promotion and application. In actual engineering projects, most of them consider macro indicators like social and economic benefits, but the focus in this system is different from the past; instead, the stabilization effect is evaluated in the perspective of the soil’s safety and food’s security. In this study, 16 indicators commonly used in the research were selected to evaluate the stabilization effect, covering five aspects: the soil’s fertility and heavy metals, crop production, the crop heavy metal accumulation and the amendment. Some items, such as the soil’s conductivity, enzymes and microbial communities, were not included in the evaluation system because the measurement methods were cumbersome and expensive and not easily evaluated. The evaluation system is feasible and can reflect more comprehensively the changes in the farmland’s quality and food security after the application of the amendment. The soil types, land use patterns and crops vary significantly in different regions of China, so the indicators are not static in the actual evaluation process but they should be focused and traded off according to local conditions and research purposes. The evaluation system of the stabilization effect is not perfect and should be improved with the development of stabilization technology.

### 4.2. Deficiencies in the Evaluation Method

Compared with previous studies [19], the comprehensive evaluation model adds a more reliable response model. Moreover, the threshold setting of each response model mainly refers to national standards of China, the classification management of farmland and previous researches, so that the evaluation score is more scientific and referential. In addition, compared with the macroscopic life cycle assessment [42], the indicators selected can better satisfy the needs of a farmland remediation project. Moreover, the weights of the indicators are determined using a combination of expert survey methods and the AHP, where experts score the importance of the indicators by means of a questionnaire. This method can clarify the importance of indicators and get the key focus part of researchers in the process of stabilization, but it is more subjective. Therefore, other methods should be introduced to increase the objectivity of the weights. The entropy weighting method and CRITIC method are based on the dispersion and mutability between the experimental measured data for a weight assignment. In the actual engineering evaluation, the joint entropy weight and CRITIC method can be introduced to correct the weights.

### 4.3. Analysis of Changes in AHM

In the laboratory application of the evaluation system, the reduction rate of the AHMs in soils with RBC and HAP was 24.9% and 47.4%, respectively, when measured on soil samples after 70 days. When the amendment for the heavy metals is at the same time as when there is adsorption and resolution behavior, after a long-term dynamic observation, the AHMs first decline, then rise and then decline again in the fluctuation of the decreasing trend; finally, they tend to reach a dynamic equilibrium, as shown in Figure 5. If only the reduction rate is used, for example, the reduction rate is 24.7% on day 40 after the addition of the HAP and the reduction rate is 58.4% on day 50, there will be a difference of 33.7% in the evaluation results. If the evaluation is done before reaching a state of smoothness, the difference in the sampling time will lead to an error result, which does not truly reflect the stabilization effect. Therefore, it is a challenge to find out when to evaluate the AHMs in order to get a more realistic evaluation effect. In view of the above problem, it is crucial to find a suitable prediction model to predict the change in the AHMs during the stabilization process. After reviewing a large amount of literature, stochastic differential equations and neural BP networks can predict the change in the available heavy metals [45], but there is little research in this area.

During the further development of soil stabilization technology, it is expected to be combined with computer techniques such as mega data and cloud computing. Additionally, a national soil database will be established to achieve an intelligent monitoring.

## 5. Conclusions

The basic design of the evaluation system for the stabilization effect of heavy metal-contaminated soil was suggested. The evaluation system consists of 1 target layer, 3 criterion layers and 16 indicator layers. According to the AHP, the available heavy metals and the heavy metals in the edible parts of crops were given higher weights in the system, even though these two indicators also represented the situation with soil pollution and food security. The indicators were split into three groups based on the evaluation standards that are currently in place for each one: indicators I with the national standards of China, indicators II with the classification management of farmland and indicators III without evaluation criteria, respectively. The evaluation criteria of the indicators were used as model thresholds to normalize the indicator scores, and then a comprehensive scoring model was used for the evaluation. The research has been provided with a theoretical foundation and data support for the evaluation of the stabilization effect.

## Figures and Tables

**Figure 1 ijerph-19-15296-f001:**
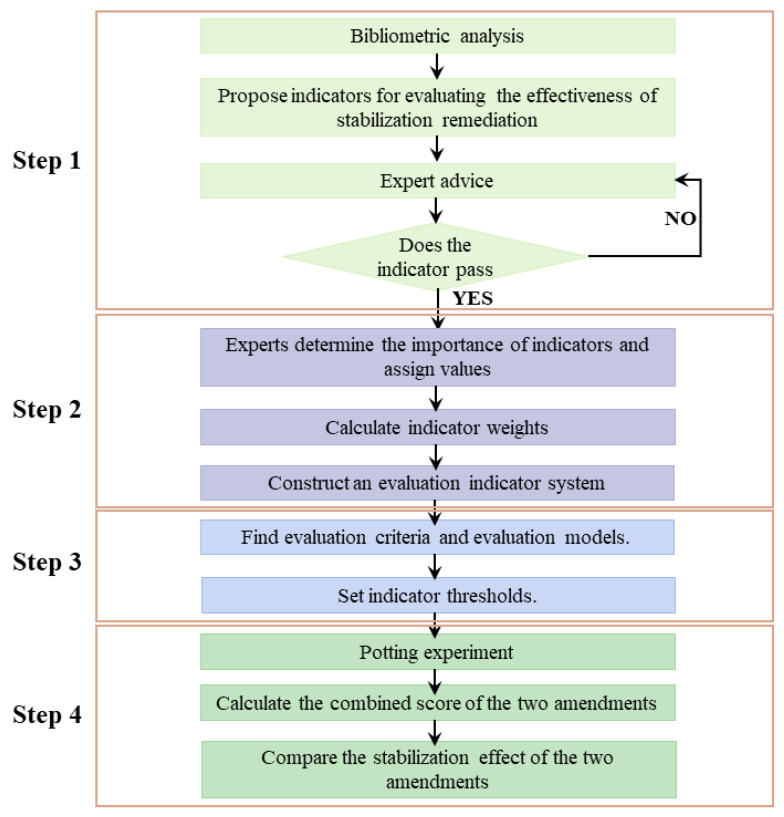
Technology Roadmap.

**Figure 2 ijerph-19-15296-f002:**
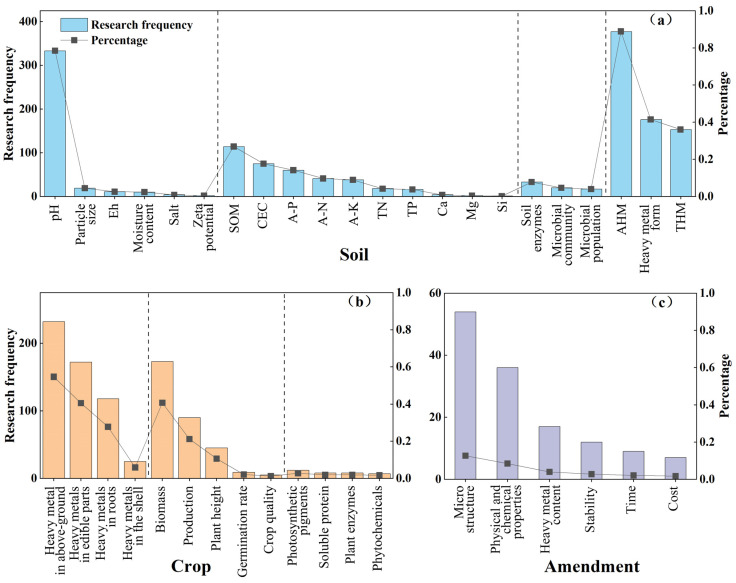
Extract of evaluation indicator of stabilization effect: (**a**) soil indicators, (**b**) crop indicators, (**c**) amendment indicators.

**Figure 3 ijerph-19-15296-f003:**
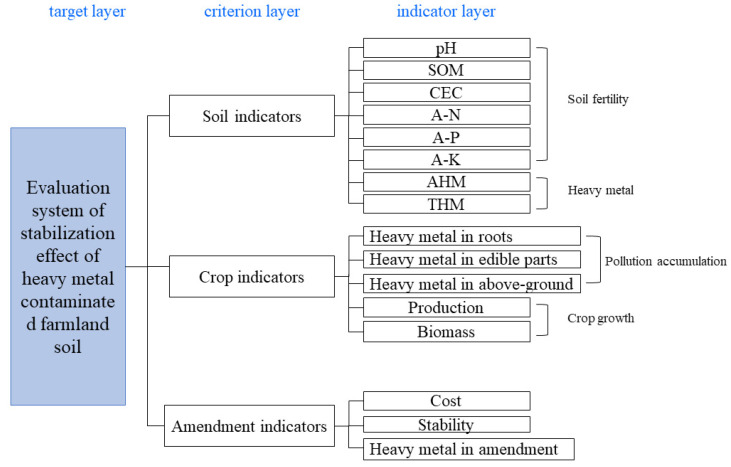
Evaluation index of the stabilization effect of heavy metal-contaminated soil.

**Figure 4 ijerph-19-15296-f004:**
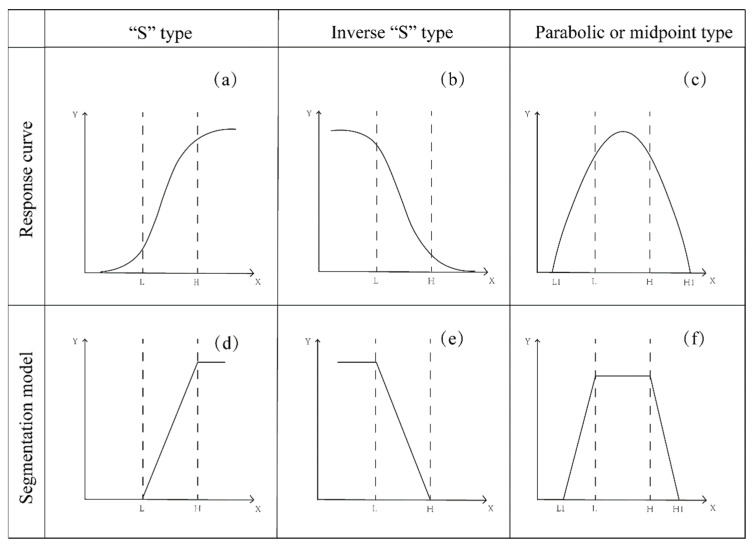
Three types of evaluation models: (**a**) “S” type response curve, (**b**) inverse “S” type response curve, (**c**) parabolic or midpoint type response curve, (**d**) “S” type segmentation model, (**e**) inverse “S” type segmentation model, (**f**) parabolic or midpoint type segmentation model.

**Figure 5 ijerph-19-15296-f005:**
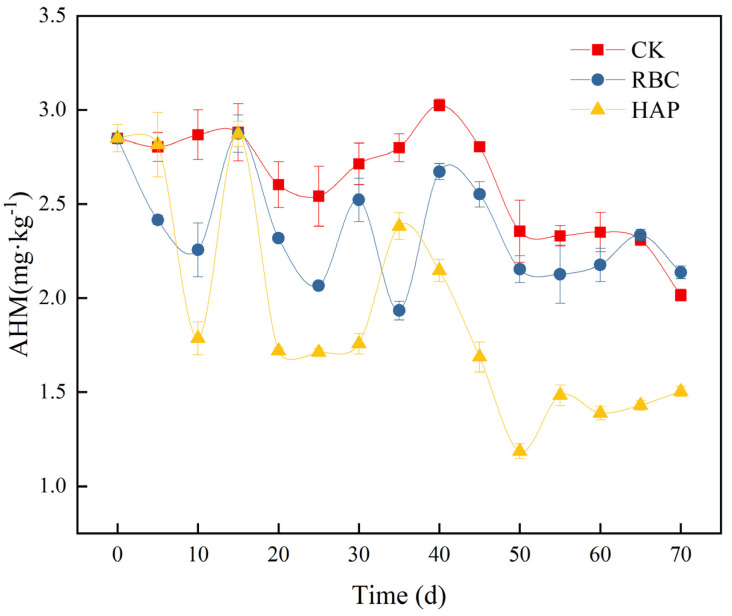
Dynamic change curve of AHM.

**Table 1 ijerph-19-15296-t001:** Analytic Hierarchy Process Scale and Relative Importance Judgment.

Element	Scale	Meaning
*b_ij_*	1	Factor *i* is of equal importance compared to factor *j*
	3	Factor *i* is slightly more important than factor *j*
	5	Factor *i* is significantly more important than factor *j*
	7	Factor *i* is strongly more important than *j* compared to factor *j*
	9	Factor *i* is extremely more important than *j* compared to factor *j*

2, 4, 6 and 8 take the middle value of the above two adjacent judgements, the reciprocal of each number from 1 to 9 has the opposite meaning to the above, e.g., 1/5 means *j* is significantly more important than *i*.

**Table 2 ijerph-19-15296-t002:** The value of the random consistency indicator RI.

*n*	1	2	3	4	5	6	7	8	9
RI	0	0	0.58	0.90	1.12	1.24	1.32	1.41	1.45

**Table 3 ijerph-19-15296-t003:** Evaluation system and indicator weight.

Target Layer	Criterion Layer	Weight	Sub-Criteria Layer	Relative Weight	Indicator Layer	Comprehensive Weight
Stabilization effect evaluation system (A)	Soil (B1)	0.544	Soil fertility (B11)	0.295	pH	0.052
					SOM	0.021
					CEC	0.017
					A-N	0.021
					A-P	0.022
					A-K	0.022
			Heavy metal (B12)	0.705	AHM	0.300
					THM	0.091
	Crop (B2)	0.316	Crop growth (B21)	0.265	Biomass	0.025
					Production	0.060
			Pollution accumulation (B22)	0.735	Heavy metal in edible parts	0.150
					Heavy metal in above-ground parts	0.028
					Heavy metal in roots	0.054
	Amendment (B3)	0.140			Cost	0.042
					Heavy metal in amendment	0.045
					Stability	0.053

The comprehensive weight is the final weight of each indicator in the indicator layer relative to system A.

**Table 4 ijerph-19-15296-t004:** The evaluation standards of Indicators I.

Indicator	Evaluation Standard
THM in soil	Soil environmental quality soil contamination risk control standards for agricultural land (for trial implementation) (GB 15618-2018)
Heavy metal in edible parts	National Food Safety Standards Limits for Contaminants in Food (GB2762-2017)
Heavy metals in amendment	Limit requirements for toxic and hazardous substances in fertilizers (GB 38400-2019)

**Table 5 ijerph-19-15296-t005:** Reference Table for Grading Evaluation Index.

Indicator	Level 1	Level 2	Level 3	Level 4	Level 5	Level 6
pH	5~7	4~5 or 7~8	3~4 or 8~9	2.5~3		
SOM(g·kg^−1^)	>40	30~40	20~30	10~20	6~10	<6
CEC(c mol (+)·kg^−1^)	>20	10~20	<10			
A-P (mg·kg^−1^)	>40	20~40	10~20	5~10	3~5	<3
A-K (mg·kg^−1^)	>200	150~200	100~150	50~100	30~50	<30
A-N (mg·kg^−1^)	>150	120~150	90~120	60~90	30~60	<30
Production	>0.9	0.8~0.9	0.7~0.8	<0.7		
Biomass	>0.8	0.7~0.8	0.5~0.7	<0.5		
Cost	<0.5	0.5~0.6	0.6~0.7	>0.7		

Production = actual production/average local production; biomass = crop weight/average local crop weight; cost = cost/crop revenue.

**Table 6 ijerph-19-15296-t006:** Different evaluation indicators and applicable response curve models.

Indicators	L	H	L_1_	H_1_	Function Type	Formula
SOM	6	40			“S” type	$Y\left(x\right)=\{\begin{array}{c} 0,\;x<L\\ \frac{x-L}{H-L},\;L\leq x\leq H\\ 1,\;x>H \end{array}$
CEC	10	20				
A-N	30	150				
A-P	3	40				
A-K	30	200				
Production		0.9				
Stability		3				
Biomass		0.8				
AHM		75%			Inverse “S” type	$Y\left(x\right)=\{\begin{array}{c} 1,x<L\\ \frac{H-x}{H-L},L\leq x\leq H\\ 0,x>H \end{array}$
THM	0.6	3.0				
Heavy metal in edible part		0.2				
Heavy metal in above-ground	10%	100%				
Heavy metal in roots	10%	100%				
Heavy metal in amendment		10				
Cost	0.5	0.7				
pH	5	7	3	9	Parabolic or midpoint type	$Y\left(X\right)=\{\begin{array}{c} \frac{x}{L_{1}},L_{1}\leq x<L\\ 1,L\leq x\leq H\\ \frac{H_{1}-x}{H_{1}-H},H<x\leq H_{1}\\ 0,X>H_{1},x<L_{1} \end{array}$

“S” type is where the indicator score is positively correlated with the indicator value within a certain range and is not affected when the indicator value is higher than the maximum value of the criterion; the inverted “S” type is the opposite of “S” type, and the indicator score is negatively correlated with the indicator value; parabolic or midpoint type is the highest score when the indicator value is within a certain range, and the score decreases when it is below or above a specific value; L and H represent the lower and higher values of the standard scoring function, with L_1_ and H_1_ representing the lowest and highest values, respectively.

**Table 7 ijerph-19-15296-t007:** Reference table for grading target level assessment scores.

Grade	I	II	III	IV	V
S	(0.8,1]	(0.6,0.8]	(0.4,0.6]	(0.2,0.4]	(0,0.2]

**Table 8 ijerph-19-15296-t008:** Reference table for evaluation of stabilization results.

S = S_b_ − S_a_	Passivation Results	Meaning
>1	Excellent	The comprehensive score increased by more than one grade after remediation
=1	Good	The comprehensive score increased by one grade after remediation
=0	Qualified	No change in comprehensive score after remediation
=−1	Poor	The comprehensive score decreased by one grade after remediation
<−1	Very poor	The comprehensive score decreased by more than one grade after remediation

**Table 9 ijerph-19-15296-t009:** The physicochemical properties of soil for pot experiment.

Physicochemical Properties		Measured Values
pH		8.026
Electrical conductivity(mS·m^−1^)		221
SOM(g·Kg^−1^)		4.469
Soil separate(mg·Kg^−1^)	<2 μm	59.88
	2~10 μm	32.11
	20~200 μm	8.01
Cd(mg·Kg^−1^)		0.18

**Table 10 ijerph-19-15296-t010:** Combined score of the two amendments.

Indicator	Control		RBC		HAP	
	Measured Values	Score	Measured Values	Score	Measured Values	Score
pH	7.27 ± 0.05	0.87	7.22 ± 0.02	0.89	7.11 ± 0.02	0.95
SOM (g·kg^−1^)	18.21 ± 1.02	0.36	21.86 ± 0.83	0.47	19.37 ± 0.76	0.39
CEC (c mol (+)·kg^−1^)	13.4 ± 0.85	0.34	13.4 ± 0.41	0.34	12.1 ± 0.32	0.21
A-N (mg·kg^−1^)	29.40 ± 4.49	0	33.92 ± 3.93	0.03	25.66 ± 5.77	0
A-P (mg·kg^−1^)	75.39 ± 25.50	1	79.01 ± 17.13	1	47.31 ± 13.13	1
A-K (mg·kg^−1^)	40.56 ± 11.13	0.06	47.78 ± 11.28	0.10	127.81 ± 12.52	0.58
AHM (mg·kg^−1^)	2.85 ± 0.07	0	2.14 ± 0.03	0.33	1.50 ± 0.03	0.63
Biomass (g)	1.16 ± 0.11	0.91	1.26 ± 0.14	0.99	1.66 ± 0.45	1
Above-ground (mg·kg^−1^)	0.60	0.98	0.40	1	0.35	1
Roots (mg·kg^−1^)	2.30	0.60	1.95	0.68	1.80	0.71
PHM (mg·kg^−1^)			0	1	0	1
Stability (a)			>3	1	>3	1
Score (S_i_)		0.165		0.371		0.471

## Data Availability

All data generated or analyzed during this study are included in this published article.

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
