# Peer review of "A Comprehensive Evaluation System for the Stabilization Effect of Heavy Metal-Contaminated Soil Based on Analytic Hierarchy Process"

_ijerph, 2022, doi:10.3390/ijerph192215296_

Round 1

Reviewer 1 Report

This is a very poorly article. 

Authors should use the word amendment rather than passivators. Please, check what passivation means.

Many times, it is not clear to me what the authors have done (see lines 80-82) as an example.

Overall, I am unable to see hypotheses, lessons learnt or many times what was done. The manuscript is chaotic (partly due to language issues).

Reviewer 2 Report

This article deals with 16 indicators  elected from three aspects of soil, crops and passivator. The 16 indicators are divided into three groups.

Introduction is clear and updated but objectives are confusing. Re-phrase last sentence of the introduction.

Methodology is well done as well as results. Conclusions are clear.

The article has an adequate size and style language.

Reviewer 3 Report

The integrated evaluation approach for soil remediation is new and can be accepted after minor revision:

1. Passivation is a special term, please define it. However, is it stabilization? According to my reading the text, stabilization seems better to describe heavy metal retained in the soils and amendments are used to stabilize them.

2. The resolution of Figure 2 is poor, please update especially x-axis in some plots.

Round 2

Reviewer 1 Report

I cannot see much improvement in the article.

It is still poorly written, including the use of etc. and many other words that should not appear in a scientific article. Word passivation is still used, for example in table 8

Line 45: Is a 1 % reduction statistically significant?

Authors mention many amendments, but only provide analysis for two of them (hydroxyapatite and modified biochar). Not clear why this is done. Why modified biochar? Moreover, the literature show many cases where hydroxyapatite addition result in more Cd availability. While hydroxyapatite can assist in the immobilization of Pb, benefits for Cd are not clear.

Line 216: Not clear how the modified biochar was prepared. Not sure about the physico-chemical characteristics  the biochar. Not sure where the soil was sampled or how to classify it. Overall, the research is not reproducible.

Lines 228 to 230: There is no explanation as to why this happens. Science implies explaining rather than just reporting observations. 
